# Neutrino Oscillations and CP Violation with the European Spallation Source Neutrino Super Beam †

**Marcos Dracos**

Institut Pluridisciplinaire Hubert Curien (IPHC), Université de Strasbourg, CNRS/IN2P3,
67037 Strasbourg, France; marcos.dracos@in2p3.fr
† Presented at the 23rd International Workshop on Neutrinos from Accelerators, Salt Lake City, UT, USA, 30–31 July 2022.

**Abstract:** The European project ESS*ν*SB, after a four-year feasibility study, has demonstrated that a neutrino facility based on the European Spallation Source and operated at the second oscillation maximum is not only compatible with the under construction neutron facility, but it also has a very high physics performance in the sector of discovery of CP violation in the leptonic sector and measurement of the CP-violating phase with high precision. This has been obtained by well optimising all parts of this neutrino facility going from the ESS proton linac up to the location of the neutrino far detector. Here, a summary of all these efforts based on the already published Conceptual Design Report is reported. A continuation of this work has recently been approved by EU. This new project includes investigations of implementation of low energy nuSTORM and ENUBET for cross-section measurements and sterile neutrino searches. Both options use mainly muons produced together with neutrinos. This "muon" orientation gives a new dimension to the project, enhancing its probability to be approved in the future.

**Keywords:** neutrino; oscillations; CPV; ESS; ESSnuSB; nuSTORM; LEnuSTORM; ENUBET; LEMNB

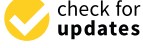



## 1. Introduction

The European Spallation Source neutrino Super Beam (ESS*ν*SB) [1] project has proposed to study the possibility of using the European Spallation Source (ESS) [2] and mainly the 5 MW proton linac, to produce the most intense neutrino beam to be used for CP violation discovery in the leptonic sector. The ESS is under construction in Lund (Sweden) since 2014 and is expected to deliver the first proton beam on target for neutron production by 2023. In a later stage, the ESS could be upgraded to double its proton beam power by doubling the proton pulse frequency in order to deliver at the same time protons for neutron and neutrino production.

The high power of the neutrino beam will allow us to operate the facility at the second neutrino oscillation maximum on which the sensitivity to CP violation is significantly higher compared to facilities operated on the first oscillation maximum [3–5]. The drawback is that the distance between the neutrino production point and the position of the far detector has to be about three times larger compared to projects operated at the first oscillation maximum using neutrinos with similar energies, thus decreasing the statistics by a factor of nine, a fact that would be prohibitive in the absence of a very powerful proton beam.

The ESS*ν*SB project has performed a Design Study in the framework of the European Union H2020 program to test the feasibility of upgrading the ESS to a neutrino facility. At the end of this study and after optimisation of all parts of the proposed facility, the physics performance to discover CP violation and precisely measure the CP violation parameter has been performed. A cost evaluation has also been conducted, included in the facility Conceptual Design Report already published [6].

In this report, a summary of all proposed ESS modifications, a description of the neutrino detectors and the reached physics performance are given.

At the end of this project, a complementary study, mainly based on measurements of neutrino cross-sections, has been submitted to the EU in the framework of Horizon Europe. This proposal has already been approved, but not yet started. A short description of this new feasibility study will also be given (Section 5).

## 2. ESS Upgrade

The ESS proton linac (Figure 1) will deliver 2 GeV proton pulses with a frequency of 14 Hz. In a previous neutrino facility preliminary feasibility report [7], it was recommended to raise the proton kinetic energy to 2.5 GeV mainly to avoid strong charge effects. This proton kinetic energy has also been adopted by ESS$\nu$SB. The linac duty cycle for neutron production will be of only 4%, leaving enough room to double it by doubling the linac frequency to go to 8% for the neutrino facility needs.

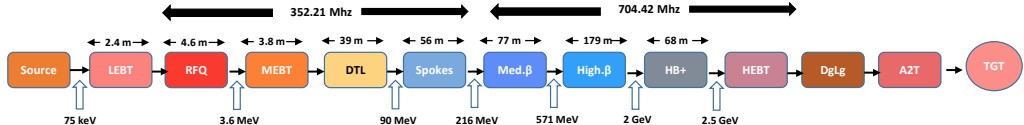

**Figure 1.** ESS proton linac.

### 2.1. Short Proton Pulses

The duration of the proton pulses for neutron users is of the order of 3 ms. This is too long for the neutrino facility because of the 350 kA current pulses to be sent to the hadronic collector (see below), necessary to produce the neutrino beam, the duration of which has to be limited to a few μs. The other reason comes from the physics requirement to keep the atmospheric neutrino background as low as possible.

In order to decrease the proton pulse duration, an accumulation ring is necessary, which circumference has to be short enough to satisfy the above requirements and also fit in the ESS allocated area, but large enough to avoid too much space charge effects due to the very high proton intensity. The chosen circumference is 385 m inducing a proton pulse duration at the exit of the accumulator of 1.2 μs.

The pulse compression is obtained through multi-turn charge-exchange injection and single-turn extraction. To be able to introduce protons in the accumulation ring while already protons are present in the ring, H$^-$ ions have to be produced and accelerated in the linac and stripped at the entrance of the ring. For this, an H$^-$ ion source is needed running in parallel with the proton one, inevitably increasing the cost of the project.

### 2.2. Neutrino Beam

The neutrino beam is produced using the classical way where the proton beam hits a target producing mesons, mainly pions, which are focused, before decaying to neutrinos and muons, in the detector direction using a magnetic horn. Due to the very high proton beam intensity, four target/horn systems will be used pulsed alternatively. To send the proton beam on the four targets a switchyard is placed at the exit of the accumulation ring as shown in Figure 2.

To produce the necessary pulsed magnetic field inside the magnetic horn focussing the mesons produced in the target located inside the horn, a pulsed current of 350 kA is necessary. A dedicated power supply needed to alternatively pulse the four horns with a frequency of 14 Hz, has been designed. To avoid irradiation of this device, it has been placed upstream of the target station, on top of the proton beam switchyard, as shown in Figure 3.

Extensive radiation studies have been performed to define the needed shielding around the target station taking into account the local safety requirements. The power dissipation and cooling requirements have also been extensively studied.

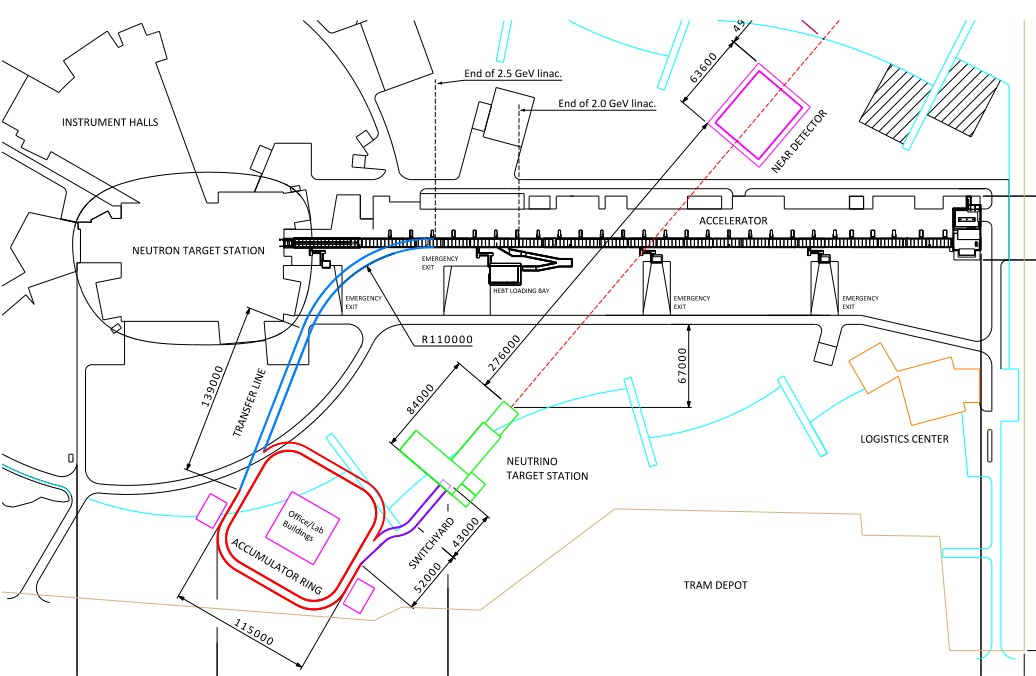

**Figure 2.** ESSνSB layout on top of the ESS facility.

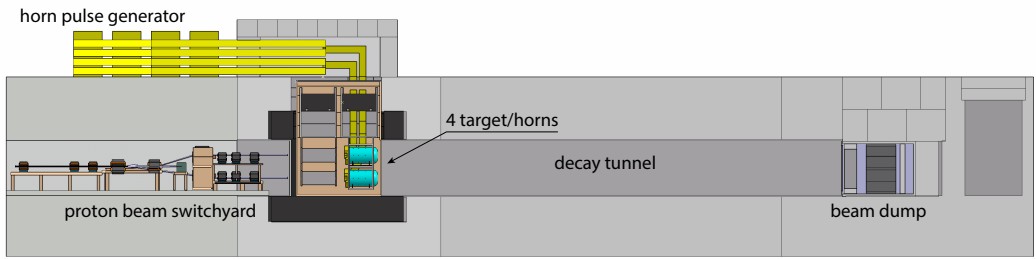

**Figure 3.** Proton switchyard, target station, decay tunnel and beam dump.

The pion decay tunnel must be long enough to leave pions decay, but short enough to avoid muons (produced together with neutrinos) to decay producing electron neutrinos, which will be a background in the $\nu_\mu \longrightarrow \nu_e$ oscillation under study. A tunnel with a length of 50 m has been chosen after many optimisations. The horn dimensions have also been optimised using a genetic algorithm.

The obtained neutrino beam at an arbitrary distance of 100 km and neglecting neutrino oscillations is depicted by Figures 4 and 5 where all neutrino energy spectra are shown for both horn current polarities, positive (neutrinos) and negative (antineutrinos). The $\nu_\mu$ purity is very high and of the order of 97%. The $\nu_e$ contamination is less than 0.5%. While this component is disturbing for CP violation measurements, it could be used by the neutrino near detector to measure the $\nu_e$ interaction cross-section at the relevant to the project energy range.

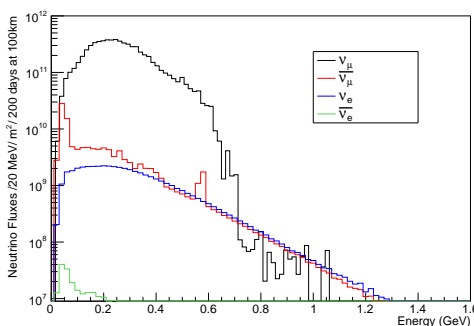

**Figure 4.** Neutrino spectrum for positive horn polarity (neutrino mode).

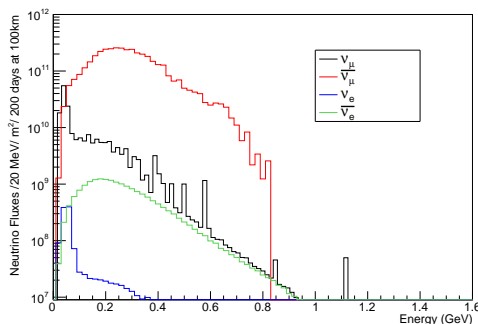

**Figure 5.** Neutrino spectrum for negative horn polarity (antineutrino mode).

## 3. Neutrino Detectors

The main goal of the near detector is to reduce the systematic uncertainties to stay below the statistical ones. This will be obtained by measuring the unoscillated neutrino flux and by measuring the neutrino interaction cross-sections at neutrino energies relevant for this project.

The neutrino near detector will be placed at a distance of 250 m from the target station (Figure 2) in the already ESS allocated area. It will be composed of a kiloton-scale Cherenkov detector (reduced size of the far detector) used for event-rate measurement and flux normalisation, neutrino interaction cross-section measurements in water and event reconstruction comparison with the far Cherenkov detector. In addition to the Cherenkov detector, a magnetised fine-grained tracker (SFGD) [8] will be used for precise tracking and energy determination. A nuclear emulsion detector for measuring precisely neutrino interaction topologies and cross-sections, similar to that of the NINJA experiment [9], will be installed upstream of the two above detectors. Figure 6 presents the whole near detector.

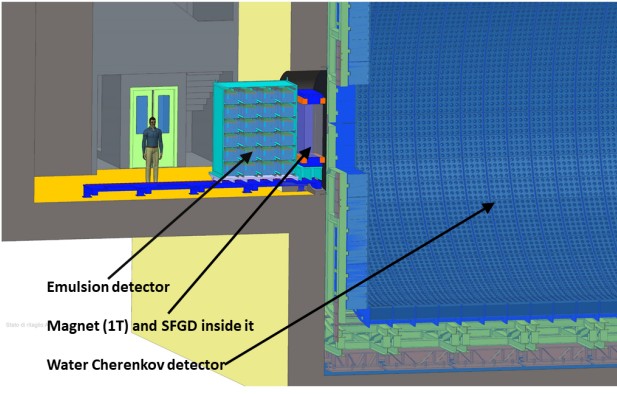

**Figure 6.** ESS$\nu$SB near detector.

For the far detector, two candidate active mines have been considered, Zinkgruvan at 360 km and Garpenberg at 540 km. After taking into account the physics performance of both choices for CP violation discovery and precision measurement of this phenomenon, the Zinkgruvan mine has been selected. The chosen detector is a SuperKamiokande-like water Cherenkov detector with a fiducial volume of around 540 kt with 20" PMTs and a photocathode coverage of 30%. Figure 7 shows one of the two far sub-detectors.

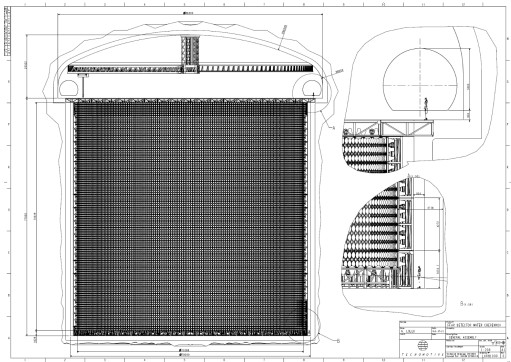

**Figure 7.** ESS$\nu$SB far water Cherenkov detector (cylinder with 78 m diameter and 78 m height).

## 4. Physics Performance

The physics performance has been evaluated considering a signal systematic uncertainty of 5%. Figure 8 presents the CP violation discovery significance versus the CP-violating phase $\delta_{CP}$ for normal mass hierarchy, for 10-years data taking. It has to be mentioned here that the proposed facility operated at the second oscillation maximum, due to the relatively short baseline, is not very sensitive to the neutrino mass hierarchy. Figure 9 shows the CP violation coverage significance versus $\delta_{CP}$ fraction. It can be seen that more than 70% of $\delta_{CP}$ values can be covered with a significance of 5 $\sigma$ after 10-years data taking. The fraction of $\delta_{CP}$ covered at 5 $\sigma$ versus time is presented by Figure 10. It can be seen that already after 5-years data taking, more than 60% of $\delta_{CP}$ values can be covered. This figure also shows that even after 20-years data taking, the results are dominated by the statistical uncertainties and not yet by systematics. In case that larger coverage is needed, it would just be enough to continue taking data without extra facility upgrades.

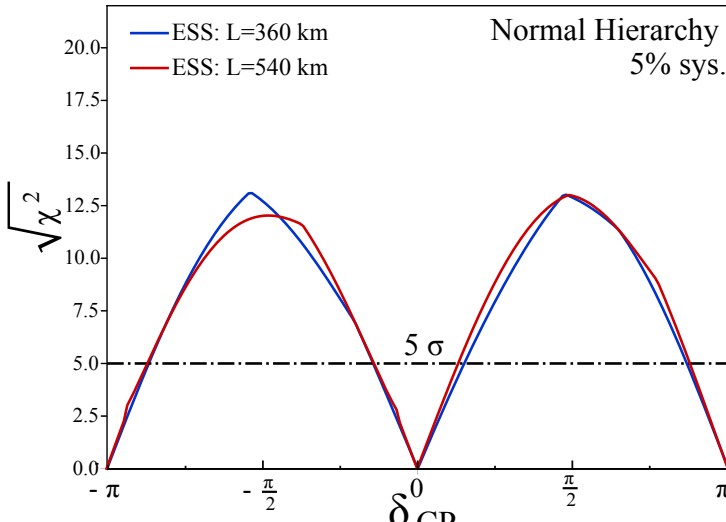

**Figure 8.** CP violation discovery significance versus $\delta_{CP}$.

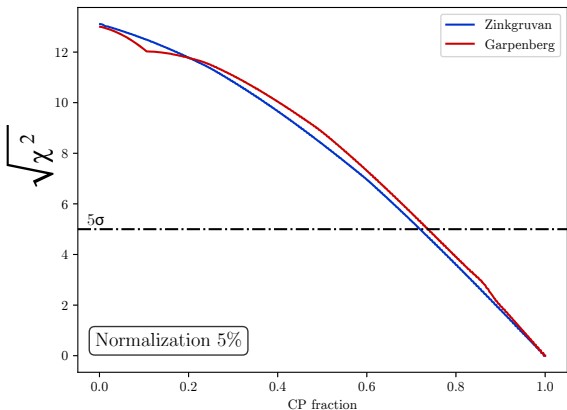

**Figure 9.** CP violation discovery significance versus $\delta_{CP}$ fraction.

After discovering CP violation, the next important step will be the precise measurement of $\delta_{CP}$. This, on top of the fact that it will help to make unitarity tests of the PMNS matrix, it would also enormously help to disregard models based on flavour symmetries predicting the $\delta_{CP}$ value (e.g., [10,11]). Figure 11 shows the expected precision on $\delta_{CP}$ versus $\delta_{CP}$. It can be seen that for the Zinkgruvan mine, for all values of $\delta_{CP}$, $\Delta\delta_{CP}$ remains below 7.5°.

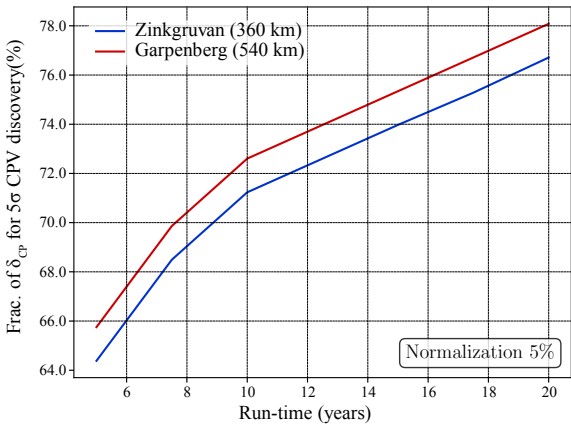

**Figure 10.** Fraction of $\delta_{CP}$ covered at 5 $\sigma$ versus data taking time.

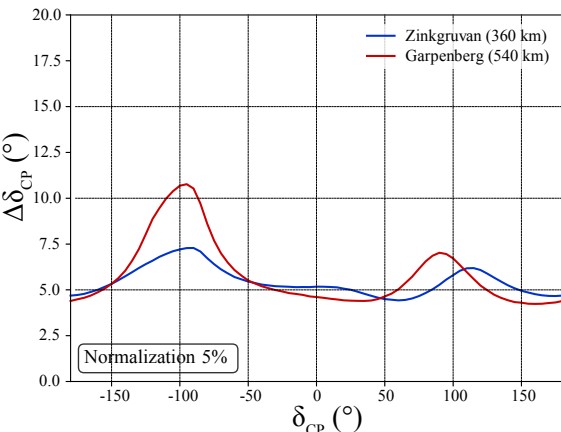

**Figure 11.** Precision on $\delta_{CP}$ versus $\delta_{CP}$.

The achieved precision result pushed the project to choose the Zinkgruvan mine (360 km) as the location of the far detector despite the fact that the Garpenberg mine (540 km) is slightly better in CP violation discovery.

## 5. Neutrino Production Using Muons

Together with the neutrino production, a copious number of muons is produced. These muons can be used for physics measurements relating to neutrino physics. Figure 12 presents the muon energy distribution for three distances from the target, 25 m, 50 m and 100 m. At 25 m, more than $4 \times 10^{21}$ are produced per year (200 operating days) with a momentum of the order of 0.5 GeV/c.

Adequately deviated from the proton beam line, these muons and not yet decayed pions can be used by a low energy nuSTORM [12] facility for neutrino cross-section measurements, muon cooling and re-acceleration R&D, and sterile neutrino searches. The pions can also be used by a low energy version of ENUBET [13], called Low Energy Monitored Neutrino Beam (LEMNB), producing a well-controlled neutrino beam. For this purpose, an instrumented decay tunnel to tag the leptons produced by pion and muon decays is used. LEMNB, as LEnuSTORM, can be used for neutrino cross-section measurements.

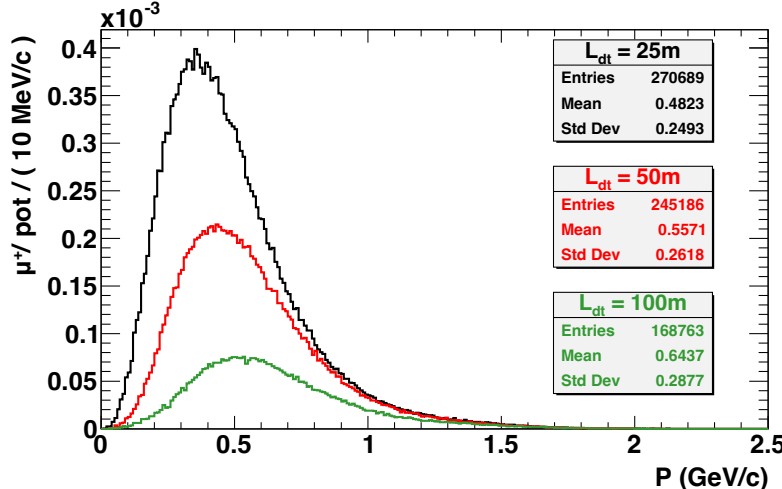

**Figure 12.** Momentum distribution for muons collected at 25 m, 50 m and 100 m from the target.

These two options have been introduced in the ESS$\nu$SB project as a possible intermediate stage before going to the CP violation searches. A new project, called ESS$\nu$SB+ [14], has been submitted to the EU in the framework of Horizon Europe. This project includes civil engineering on ESS and far detector sites, the design of a reduced proton power target station for LEnuSTORM and LEMNB and a detector design. Figure 13 depicts the layout of the new ESS$\nu$SB+ project together with those of ESS$\nu$SB. The same detector will be used by LEMNB and as near detector of LEnuSTORM. The LEnuSTORM far detector will be the same as the one to be used after by ESS$\nu$SB for CP violation searches.

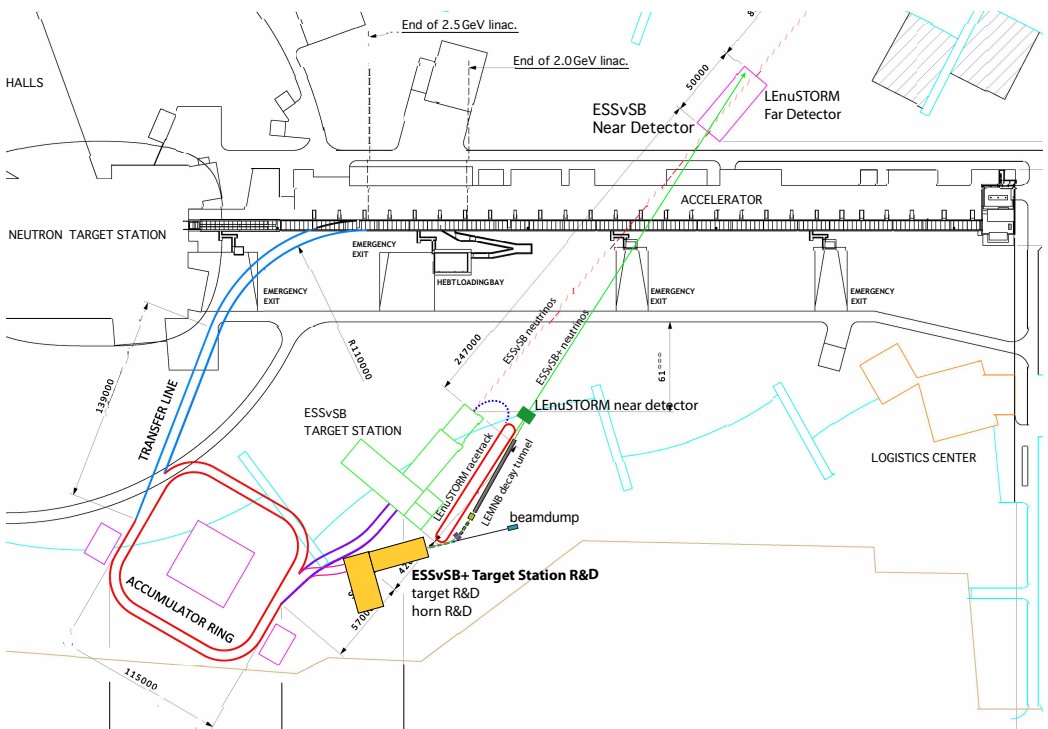

**Figure 13.** Layout of ESSνSB+ facility with the "low power" target station (yellow) followed by the low energy nuSTORM ring (red) and LEMNB (low energy ENUBET) decay tunnel.

## 6. Conclusions

The European Spallation Source neutrino Super Beam Design Study has been financed by EU under the framework of Horizon 2020. The Conceptual Design Report, now published, summarises all studies conducted during four years between 2018 and 2022. Not only ESSνSB proved the feasibility of the production of a very intense neutrino beam using the 5 MW ESS proton beam, but also that the physics performance obtained, operating the facility at the second oscillation maximum, surpasses all expectations. Indeed, in ten years data taking, this project can reach, for CP violation discovery, a $\delta_{CP}$ coverage of more than 70%. The project can also reach a $\delta_{CP}$ precision lower than 7.5° for all $\delta_{CP}$ values.

Strong of this success, a second project, ESSνBS+, has been submitted to EU under the Horizon Europe framework, with the aim to even further enhance the possibilities and potentiality of ESSνBS. This second project now accepted by EU includes the feasibility study of adding to the project a low energy nuSTORM for neutrino interaction cross-section measurements and sterile neutrino searches. It also includes a low energy ENUBET study for neutrino cross-section measurements.

**Funding:** This research was funded by the European Union's Horizon 2020 research and innovation programme under grant agreement No 777419.

**Institutional Review Board Statement:** Not applicable.

**Informed Consent Statement:** Not applicable.

**Data Availability Statement:** Not applicable.

**Conflicts of Interest:** The author declares no conflict of interest.

## Abbreviations

The following abbreviations are used in this manuscript:

| | |
|---|---|
| ESS | European Spallation Source |
| ESSνSB | European Spallation Source neutrino Super Beam |
| CPV | Charge–Parity symmetry Violation |
| nuSTORM | neutrinos from Stored Muons |
| LEnuSTORM | Low Energy neutrinos from Stored Muons |
| ENUBET | Enhanced NeUtrino BEams from kaon Tagging |
| LEMNB | Low Energy Monitored Neutrino Beam |
| SFGD | Super Fine Grain Detector |
| NINJA | Neutrino Interaction research with Nuclear emulsion and J–PARC Accelerator |
| PMT | Photo–Multiplier Tube |
| PMNS | Pontecorvo–Maki–Nakagawa–Sakata (mixing matrix) |

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
