# Peer review of "Neutrino Oscillations and CP Violation with the European Spallation Source Neutrino Super Beamâ€"

_psf, doi:10.3390/psf8010067_

Round 1
Reviewer 1 Report
Comments and Suggestions for Authors
The manuscript provides a good overview of the ESSnSB project and its future prospects. In general, the proceeding is clearly written and is of high interest for the neutrino community, also showing the great potential of this experiment. Therefore, I recommend its publication after considering some questions and suggestions that can be found below these lines and that could make the article more readable and self-contained.
Comments and suggestions:
- At the beginning of page 2, it is stated that a short description of neutrino cross-section studies will also be given. If appropriate, it could be specified in which section or sections these new studies will be described.
- In later phases of the experiment, MC event generators would be necessary. Could the author add any information about this matter? Which generator will be used or which ones are being considered?
- Typo in line 131: projet --> project
- Is there any specific reference to the new ESSnuSB+ project? If so, please include it in the manuscript.
Author Response
- At the beginning of page 2, it is stated that a short description of neutrino cross-section studies will also be given. If appropriate, it could be specified in which section or sections these new studies will be described.
Done (Sec. 5)
- In later phases of the experiment, MC event generators would be necessary. Could the author add any information about this matter? Which generator will be used or which ones are being considered?
For all studies GENIE has been used. I don't think at this level it is necessary to give this kind of details.
- Typo in line 131: projet --> project
Done
- Is there any specific reference to the new ESSnuSB+ project? If so, please include it in the manuscript.
Not yet, there is an EU page, now included in references.